# Caffeine Supplementation and Physical Performance, Muscle Damage and Perception of Fatigue in Soccer Players: A Systematic Review

**DOI:** 10.3390/nu11020440

**Published:** 2019-02-20

**Authors:** Juan Mielgo-Ayuso, Julio Calleja-Gonzalez, Juan Del Coso, Aritz Urdampilleta, Patxi León-Guereño, Diego Fernández-Lázaro

**Affiliations:** 1Department of Biochemistry and Physiology, School of Physical Therapy, University of Valladolid, 42004 Soria, Spain; 2Laboratory of Human Performance, Department of Physical Education and Sport, Faculty of Physical Activity and Sport, University of the Basque Country, 01007 Vitoria, Spain; julio.calleja.gonzalez@gmail.com; 3Exercise Physiology Laboratory, Camilo José Cela University, 28692 Madrid, Spain; jdelcoso@ucjc.edu; 4Elikaesport, Nutrition, Innovation & Sport, 08290 Barcelona, Spain; a.urdampilleta@drurdampilleta.com; 5Faculty of Psychology and Education, University of Deusto, Campus of Donostia-San Sebastián, 20012 San Sebastián, Guipúzcoa, Spain; patxi.leon@deusto.es; 6Department of Cellular Biology, Histology and Pharmacology. Faculty of Physical Therapy, University of Valladolid. Campus de Soria, 42004 Soria, Spain; diego.fernandez.lazaro@uva.es

**Keywords:** football, RPE, DOMS, sport performance, supplementation, ergogenic aids

## Abstract

Soccer is a complex team sport and success in this discipline depends on different factors such as physical fitness, player technique and team tactics, among others. In the last few years, several studies have described the impact of caffeine intake on soccer physical performance, but the results of these investigations have not been properly reviewed and summarized. The main objective of this review was to evaluate critically the effectiveness of a moderate dose of caffeine on soccer physical performance. A structured search was carried out following the Preferred Reporting Items for Systematic Review and Meta-Analyses (PRISMA) guidelines in the Medline/PubMed and Web of Science databases from January 2007 to November 2018. The search included studies with a cross-over and randomized experimental design in which the intake of caffeine (either from caffeinated drinks or pills) was compared to an identical placebo situation. There were no filters applied to the soccer players’ level, gender or age. This review included 17 articles that investigated the effects of caffeine on soccer-specific abilities (*n* = 12) or on muscle damage (*n* = 5). The review concluded that 5 investigations (100% of the number of investigations on this topic) had found ergogenic effects of caffeine on jump performance, 4 (100%) on repeated sprint ability and 2 (100%) on running distance during a simulated soccer game. However, only 1 investigation (25%) found as an effect of caffeine to increase serum markers of muscle damage, while no investigation reported an effect of caffeine to reduce perceived fatigue after soccer practice. In conclusion, a single and moderate dose of caffeine, ingested 5–60 min before a soccer practice, might produce valuable improvements in certain abilities related to enhanced soccer physical performance. However, caffeine does not seem to cause increased markers of muscle damage or changes in perceived exertion during soccer practice.

## 1. Introduction

Soccer is considered one of the most popular sports worldwide. According to the Fédération Internationale de Football Association (FIFA) Big Count survey, there are 265 million active soccer players and the number is progressively increasing, especially in women’s football [1]. In addition, soccer attracts millions of television spectators while the socio-economic impact of elite soccer affects almost every culture worldwide [2]. Thus, the study of soccer and the variables that affect performance in this complex team sport can have a great impact on sport sciences. Briefly, modern soccer is characterized by the continuous combination of short sprints, rapid accelerations/decelerations and changes of direction interspersed with jumping, kicking, tackling and informal times for recovery [3]. In addition to these physical fitness variables, players’ techniques and cognitive capacity, team tactics, and psychological factors might also have an impact on overall soccer performance [4,5]. Unlike other team sports, such as basketball or handball, soccer is a low-scoring game and, thus, the margins of victory are close/reduced, particularly at the elite level. In consequence, the study of the effects of ergogenic aids on performance have become an important subject for players, coaches and sport scientists associated with soccer because it has the potential to increase success in the game [6]. 

Caffeine (1, 3, 7-trimethylxanthine) is one of the most popular supplements among athletes for its potent stimulant effects and due to its easy availability in the market in different commercial forms (energy drinks, caffeinated beverages, pills, pre-workout and thermogenic supplements, etc.). In addition, the ergogenic effects of the acute ingestion of caffeine have been widely reported on different forms of exercise, although most of the classic studies focused on endurance performance [7,8]. In the last few years, several investigations have found that caffeine can also increase anaerobic and sprint performance, although the direct application of these research outcomes to the complexity of soccer is complicated [9,10,11]. According to the Australian Institute of Sport (AIS), the potential ergogenicity of caffeine reflects level 1 evidence, which allocates it as a safe supplement to use in sport [12]. In addition, the International Olympic Committee indicates, in its recent consensus statement for dietary supplements, that caffeine intake results in performance gains when ingested before exercise in doses ranging from 3 to 6 mg/kg. Finally, two recent systematic reviews have concluded that caffeine might be ergogenic in team sport athletes [6,13]. With this background, one might suppose that caffeine is also ergogenic in soccer although the information regarding this sport has not been summarized. In the last few years, several studies have investigated the effects of caffeine intake on soccer physical performance [14,15,16,17,18,19,20,21] and in the opinion of the authors, the results of these investigations need to be objectively reviewed and summarized. Therefore, the objective of this systematic review was to critically evaluate the effectiveness of a moderate dose of caffeine on soccer physical performance, muscle damage and perception of fatigue in order to provide more objective and comprehensive information about the positive and negative impact of caffeine on soccer players.

## 2. Methods

### 2.1. Search Strategies

The present article is a systematic review focusing on the impact of caffeine intake on soccer physical performance and it was conducted following the Preferred Reporting Items for Systematic Review and Meta-Analyses (PRISMA) guidelines and the PICOS model for the definition of the inclusion criteria: P (Population): “soccer players”, I (Intervention): “impact of caffeine on soccer physical performance, muscle damage and perception of fatigue”, C (Comparators): “same conditions with placebo”, O (Outcome): “soccer-specific abilities, serum markers of muscle damage and perceived fatigue (RPE) and heart rate”, and S (study design): “double-blind and randomized cross-over design” [22]. 

A structured search was carried out in the Medline (PubMed) database and in the Web of Science (WOS) which includes other databases such as BCI, BIOSIS, CCC, DIIDW, INSPEC, KJD, MEDLINE, RSCI, SCIELO, both high quality databases which guarantee good bibliographic support. The search covered from July 2006, when Hespel et al., [23] suggested the use of caffeine as an effective supplement for soccer athletic performance, to November 2018. Search terms included a mix of Medical Subject Headings (MeSH) and free-text words for key concepts related to caffeine and soccer physical performance, muscle damage or perceived fatigue as follows: (“football”(All Fields) OR “soccer”(All Fields)) AND (“caffeine”(All Fields) OR “energy drink”(All Fields)) AND ((“physical performance”(All Fields) OR performance(All Fields))) OR ((“muscles”(MeSH Terms) OR “muscles” (All Fields) OR “muscle” (All Fields))) OR damage(All Fields) OR (RPE(All Fields) OR “perceived fatigue “(All Fields)). Through this search, relevant articles in the field were obtained applying the snowball strategy. All titles and abstracts from the search were cross-referenced to identify duplicates and any potential missing studies. Titles and abstracts were then screened for a subsequent full-text review. The search for published studies was independently performed by two authors (JMA and JCG) and disagreements about physical parameters were resolved through discussion.

### 2.2. Inclusion and Exclusion Criteria

For the articles obtained in the search, the following inclusion criteria were applied to select studies: articles (1) depicting a well-designed experiment that included the ingestion of an acute dose of caffeine—or a caffeine-containing product—before and/or during exercise in humans; (2) with an identical experimental situation related to the ingestion of a placebo performed on a different day; (3) testing the effects of caffeine on soccer-specific tests and/or real or simulated matches; (4) with a double-blind, and randomized cross-over design; (5) with clear information regarding the administration of caffeine (relative dose of caffeine per kg of body mass and/or absolute dose of caffeine with information about body mass; timing of caffeine intake before the onset of performance measurements, etc.); (6) where caffeine was administered in the form of a beverage, coffee gum or pills; (7) on soccer players with previous training backgrounds in soccer; (8) the languages were restricted to English, German, French, Italian, Spanish and Portuguese. The following exclusion criteria were applied to the experimental protocols of the investigation: (1) the use of caffeine doses below 1 mg/kg or above 9 mg/kg; (2) the absence of a true placebo condition; (3) the absence of pre-experimental standardizations such as elimination of dietary sources of caffeine 24 h before testing; (4) carried out in participants with a previous condition or injury. There were no filters applied to the soccer players’ level, sex or age to increase the power of the analysis. Moreover, the Physiotherapy Evidence Database scale (PEDro), the key factors of which assess eligibility criteria, random allocation, baseline values, success of the blinding procedures, power of the key outcomes, correct statistical analysis and measurement of participants’ distribution of studies, was used to evaluate whether the selected randomized controlled trials were scientifically sound: 9–10 = excellent, 6–8 = good, 4–5 = fair, and <4 = poor) [24]. Papers with a poor PEDro score were excluded (i.e., <4 points).

Once the inclusion/exclusion criteria were applied to each study, data on study source (including authors and year of publication), study design, caffeine administration (dose and timing), sample size, characteristics of the participants (level and sex), and final outcomes of the interventions were extracted independently by two authors (JMA and JCG) using a spreadsheet (Microsoft Inc, Seattle, WA, USA). Subsequently, disagreements were resolved through discussion until a consensus was reached. Experiments were clustered by the type of test used to assess the effects of caffeine on soccer physical performance and groups of experiments were created on the effects of caffeine physical performance, muscle damage and perception of fatigue because of its importance to overall soccer performance [4].

## 3. Results

### 3.1. Main Search

The literature search provided a total of 135 articles related to the selected descriptors, but only 17 articles met all the inclusion/exclusion criteria (see Figure 1). The number of articles and their exclusion criteria were: 32 papers were removed because they were duplicated; 4 papers were removed because they were performed on a non-human population; another 4 papers were removed because they were narrative or systematic reviews; 13 studies were not carried out during the range of dates included in the inclusion criteria. From the remaining 40 articles, another 23 papers were removed because they were unrelated to the effects of caffeine on soccer physical performance. The topics and number of studies that were excluded were: 1 because of lack of information on body mass, 1 because the caffeine content was found in nutritional supplements with other drugs, 1 because it was a suggestion for future research, 1 because the sport investigated was not specified, 4 because they dealt with recovery or sleep processes, 4 because they investigated other team sports (1 rugby, 1 volleyball, 1 tennis, 1 Gaelic football), and the remaining 11 articles because they studied other subjects unrelated to the focus of this systematic review. Thus, the current systematic review includes 17 studies. 

### 3.2. Caffeine Supplementation

The participants’ samples included players of both genders (241 males and 33 females), who competed in professional or elite (*n* = 108), semi-professional (*n* = 19) and amateur teams (*n* = 147). In addition, 70 players were adolescents. Out of the 17 investigations, only 2 studies included female soccer players. In 12 out of 17 studies, caffeine was administered based on the soccer player’s body mass, while an absolute dose was provided for all participants in 5 studies. In 2 studies the caffeine dose employed was less than 3 mg/kg, 3 studies used a caffeine dose of around 3 mg/kg, in 2 studies it was 4.5 mg/kg, in 3 studies it was around 5 mg/kg, in 4 studies the dose was 6 mg/kg and 2 studies included a dose above 6 mg/kg (i.e., 7.2 mg/kg). In 1 study, soccer players took different doses (1, 2 and 3 mg/kg). Regarding the form of administration, 9 investigations used capsules filled with caffeine, 3 investigations used caffeinated energy drinks, 3 investigations used a caffeinated sport drink, 1 investigation employed a 20% carbohydrate solution and 1 investigation employed caffeinated chewing gum. 

Most investigations administered caffeine 30–60 min prior to testing, with the exception of the studies conducted by Andrade-Souza et al. (2015) where the consumption of caffeine was carried out 3 h after a practice session, 4 h after its effects were evaluated in a simulated match [25]. Also, Guttierres et al. (2013) used a protocol that included the ingestion of caffeine 1 h before the test and every 15 min during the protocol [26]. Finally, Ranchordas et al. (2018) employed caffeine 5 min before the tests because they used caffeinated gums [17]. In summary, different studies examined the effect of caffeine on soccer physical performance by using a variety of times of ingestion prior to the testing (5 min–60 min).

### 3.3. Outcome Measures

Table 1, Table 2 and Table 3 include information about author/s and year of publication; the sample investigated, with details of sport level, sex and the number of participants; the study design cites the control group if the study included one; the supplementation protocol that specifies the type of caffeine used, the dose and the time that it was administered; the parameters analyzed or main effects either on sport performance (*n* = 12; Table 1) and muscle damage (*n* = 5; Table 2) and finally results or main conclusions. Additionally, some studies also presented data on the effects of caffeine on perceived exertion and heart rate (*n* = 6; Table 3).

## 4. Discussion

The purpose of this systematic review was to summarize all scientific evidence for the effect of acute caffeine ingestion on variables related to soccer physical performance. Due to the differences of the effects studied among the investigations included in the analysis, the following variables have been clustered for a more comprehensive scrutiny.

### 4.1. Impact on Sports Performance

A total of 12 investigations carried out research protocols that studied the effects of caffeine on one or more variables related to soccer-specific abilities. Overall, these investigations showed an improvement in soccer-related skills with the pre-exercise ingestion of caffeine (Table 1). Specifically, Foskett et al., [15], with 12 first division football players (age: 23.8 ± 4.5 years), observed that the consumption of 6 mg/kg of caffeine before exercise increased passing accuracy and accrued significantly less penalty time during two validated tests to assess soccer skill performance (intermittent shuttle-running protocol and Loughborough Soccer Passing Test; LSPT). In addition, this investigation also found that caffeine improved the functional power of the leg measured by a vertical jump. In the study conducted by Jordan et al., 17 soccer players from the elite youth category (age: 14.1 ± 0.5 years) performed an agility test (reactive agility test) validated for football [34]. These authors indicated, based on the results of their investigation, the intake of 6 mg/kg of caffeine 60 min before the test significantly improved the reaction time of the players in their non-dominant leg [16]. In another study conducted with 15 elite young players (age: 16 ± 1 years) that were administered low doses of caffeine (1, 2 and 3 mg/kg), Ellis et al., [21] observed that improvements in physical performance depended on the dose and the type of task. Specifically, they concluded that 3 mg/kg of caffeine seems to be the optimal dose to obtain positive effects on soccer-specific tests (20 m sprint, arrowhead agility and CMJ). However, the authors also suggested that even higher doses of caffeine might be required to improve endurance performance, as measured by the Yo-Yo intermittent recovery test level 1 (Yo-Yo IR1).

In this line, Apostolidis et al., [19] showed that 6 mg/kg of caffeine ingested 60 min previous to a battery of tests improved aerobic endurance (time to fatigue) and neuromuscular performance (CMJ) in 20 well-trained soccer players (age: 21.5 ± 4 years). Since these authors did not find any change in substrate oxidation with caffeine, measured by indirect calorimetry during the testing, they commented that performance improvements could only be attributed to positive effects on the central nervous system and/or neuromuscular function, although the precise mechanism of caffeine ergogenicity was not indicated in this investigation. Finally, Guerra et al., [18] investigated the addition of caffeine (5 mg/kg) to a post-activation potentiation protocol that included plyometrics and sled towing. These authors found that, in a group of 12 male professional soccer players (age: 23 ± 5 years), caffeine augmented the effects of the post-activation potentiation, as measured by CMJ. These investigations, taken together, suggest that caffeine might be effective to improve performance in players’ abilities and soccer-specific skills (jumps, sprint, agility, aerobic endurance, accuracy of passes and ball control). 

Caffeinated energy drinks are considered as one of the most common ways to provide caffeine before exercise [14], and the effect of this type of beverages have been also investigated in soccer players. Del Coso et al., [16] chose 19 semi-professional players (age: 21 ± 2 years) in order to determine if the caffeine, provided via a commercially-available energy drink (3 mg/kg), improved performance during several soccer-specific tests (single and repeated jump tests and repeated sprint ability test) and during a simulated soccer match. For this investigation, players ingested either an energy drink without sugar but with caffeine (i.e., sugar-free Redbull), or a sugar-free soda (Pepsi diet without caffeine) 60 min prior to testing. The results showed that the consumption of the caffeinated energy drink increased the ability to jump, to repeat sprints, and it affected positively total running distance and the running distance at >13 km/h covered during the simulated game. In another similar study carried out with 18 semi-professional women soccer players (age: 21 ± 2 years), Lara et al., [34] demonstrated that the consumption of an energy drink containing 3 mg/kg of caffeine improved jump height, the ability to perform sprints, the total running distance and the distance covered at high running intensity (i.e., >18 km/h). These two investigations together suggest that caffeine in the form of an energy drink, can improve the physical demands associated with high performance in soccer, such as sprints, rapid accelerations/decelerations and constant changes of direction [35,36]. 

Thanks to the collaboration of 18 junior soccer players (age: 16.1 ± 0.7 years), Guttierres et al., [29] evaluated whether the physical performance of these players increased with the consumption of a caffeinated sports drink (250 mg/L ≈ 7.2 mg/kg) compared to a commercial carbonated drink. They concluded that the caffeine-based sport drink significantly increased jump height and improved the power in the lower limbs, another determinant factor in soccer physical performance [37]. However, positive effects were not demonstrated in the “Illinois Agility Test”, a validated routine to assess agility in team sports players [38]. This lack of positive effects was possibly due to the fact that players’ agility is a complex process that depends on the coordination of factors such as decision making and speed in the changes of direction, both aspects of which are trained and constantly improved in training [39]. Gant et al., [28] used a caffeine-containing carbonated drink in 15 professional soccer players (age: 21.3 ± 3 years) and were able to evidence that the addition of caffeine to a carbonated drink, in a dose of 3.7 mg/kg (60 min before starting and every 15 min during the test), improved sprint performance and the vertical jump in soccer players. Andrade-Souza et al., [25] proposed a very interesting study where the aim was to investigate the effect of a carbohydrate-based drink, with (6 mg/kg) and without caffeine, and they compared these trials to the isolated ingestion of caffeine. The study was carried out with 11 college football players (age: 25.4 ± 2.3 years) with the ingestion of the drinks after the morning training session to see the effects of the 20% carbohydrate solutions on the afternoon training session of the same day. The main finding of Andrade-Souza’s study was that none of the drinks was able to increase performance. This fact could be related to reduction of the glycogen levels from the morning to the afternoon training sessions, a factor that was either not considered or measured in the study. According to Jacobs [40], a reduction in the content of muscle glycogen below a critical threshold can affect anaerobic strength and performance. In the Andrade-Souza’s study, the recovery time between the two practice sessions was very short (4 h) and the nutritional strategies chosen were not optimal to replenish muscle glycogen [25]. Thus, it is likely that the reduction of glycogen stores might have precluded the ergogenic effect of caffeine in this experimental design. 

Chewing gum provides an alternative mode of caffeine administration that is more rapidly absorbed (via the buccal mucosa) than capsules and drinks (i.e., 5 min vs. 45 min, respectively) and less likely to cause gastrointestinal distress [41]. Along these lines, Ranchordas et al., reported that caffeinated chewing gum, containing 200 mg of caffeine, can enhance aerobic capacity (Yo-Yo IR1) by 2% and increase CMJ performance by 2.2% in 10 male university soccer players (age: 19 ± 1 years) [17]. Therefore, chewing gum could be beneficial for soccer players where the time between ingestion and performance is short, (e.g., for substitutes that would come on when called upon by the coach and for players who cannot tolerate caffeinated beverages or capsules because of gastrointestinal distress before kick-off [17]).

### 4.2. Impact on Muscle Damage

A total of 5 investigations studied the effect of caffeine on the levels of muscle damage after a soccer practice (Table 2). In soccer, muscle damage is a very important physiological variable because all the high-intensity exercises produced in this sport (sprints, accelerations/decelerations, changes of directions and even tackles) may be associated with myofibril damage [42]. In this way, Machado et al. carried out three studies on this specific topic: one with 20 healthy soccer players (age: 18.8 ± 1 years) [31]; another one with 15 male soccer players (age: 19 ± 1 years) [32] and the last one with 15 male soccer athletes (age: 18.4 ± 0.8 years) [30]. The main goal of these three investigations was to determine whether consumption of caffeine in single and acute doses (4.5–5.5 mg/kg) negatively affected blood markers typically used to assess the level of muscle damage. The authors concluded that these markers (creatine kinase, lactate dehydrogenase, aspartate aminotransferase and alanine aminotransferase) increased with exercise, but they did not find that this increase was exacerbated with the consumption of caffeine. On the other hand, a study conducted by Bassini-Cameron et al., [33] which measured hematological variables, muscle proteins and liver enzymes in 22 professional soccer players (age: 26.0 ± 1.6 years), concluded that 5 mg/kg of caffeine ingested 60 min before the start of a game increased the risk of muscle damage in players because there was an increase in the white blood cell count. However, the serum concentration of white blood cells is not a definitive marker of muscle damage level during exercise, as other factors can increase the count of leukocytes during exercise. In summary, although most of studies found an absence of effect of caffeine on exercise-induced muscle damage in soccer players, further research is necessary to confirm this notion [43].

Guttierres et al., [26], in an experiment with 20 youth soccer players (age: 16.1 ± 0.7 years), observed the effect of caffeine (contained in a sport drinks) on free fatty acids mobilization. Participants consumed the beverage 20 min before a soccer match and every 15 min during the game with a total caffeine consumption of 7.2 mg/kg. The authors certified that the caffeine did not increase the mobilization of free fatty acids. Anyway, the caffeinated beverage was also rich in carbohydrates, increasing blood glucose concentration, promoting more insulin and therefore inhibiting the mobilization of fatty acids. Graham, et al., [44] showed that caffeine with glucose increased insulin secretion versus glucose consumption alone. In the Guttierres study, blood glucose concentrations were higher possibly due to an increase in sympathetic nervous system activity [26], increasing adrenaline and noradrenaline and glycogenolysis [45,46]. Moreover, blood lactate concentrations increased with the ingestion of caffeine, likely indicating, in an indirect manner, that players achieved a greater intensity in the trial with caffeine. These outcomes could suggest that soccer players exercised at a higher percentage of their maximum heart rate (caffeine: 80.6 % versus control: 74.7% of maximum heart rate) with the ingestion of caffeine. 

### 4.3. Impact on the Perception of Fatigue

Astorino et al., [20] concluded that the consumption of a serving portion of an energy drink (Redbull), with 1.3 mg/kg of caffeine, did not alter the perception of fatigue or heart rate in 15 semi-professional soccer players (age: 19.5 ± 1.1 years; Table 3). An important limitation of this study was that the amount of caffeine consumed was very low (80 mg) and thus, the effects of caffeine would have been limited due to dosage. On the contrary, Guttierres et al., [26] and Lara et al., [28] found that caffeine tended to cause an increase in exercise heart rate with a concomitant reduction in the perception of fatigue. While caffeine has been proven to reduce perception of fatigue in exercise protocols that used a fixed exercise intensity [47], these investigations [20,26,28] used exercise protocols more applicable to soccer, where exercise intensity can be freely chosen, as happens during soccer play. Under these specific conditions, caffeine served to increase exercise intensity (as indicated by higher running distances and higher average heart rate) while perception of fatigue was unaffected or tended to be reduced. Thus, it might be speculated that caffeine might have the capacity to enhance soccer physical performance without producing higher values of fatigue, which can be understood as a positive property of this stimulant. 

### 4.4. Caffeine Dose and Inter-Individual Responses to Caffeine Administration

The dose of caffeine administered in the experiments ranged from 1.3 to ~7.2 mg/kg and thus, the ergogenic effects of caffeine on soccer players must be attributed to this range of dosage. However, it is still possible that dose-response effects exist or even that a caffeine dose threshold is necessary to obtain benefits from caffeine in soccer, as recently suggested by Chia et al., for ball sports [6]. Based on previous investigations with other different forms of exercise [48,49,50], or soccer [20,21], it can be suggested that doses below 2 mg of caffeine per kg of body mass might not be effective to increase soccer physical performance. All the experiments included in this systematic review reported their findings as a group mean comparing caffeine vs. placebo trials. Nevertheless, recent investigations have shown that not all individuals experience enhanced physical performance after the ingestion of moderate doses of caffeine [51,52,53,54]. These studies have identified the presence of athletes who obtain minimal ergogenic effects or only slight ergolytic effects after acute administration of caffeine, and such participants have been catalogued as “non-responders to caffeine” [55]. To date, there is still no clear explanation for the lack of ergogenic effects after the acute administration of caffeine in some individuals, although factors such as training status, habitual daily caffeine intake, tolerance to caffeine, and genotype variation have been proposed as possible modifying factors for the ergogenicity of caffeine [55]. Whilst this systematic review suggests that the ingestion of 3–6 mg/kg of caffeine is ergogenic for soccer players, it might not be optimal for everyone. The inter-individual variability in the ergogenic response to acute caffeine ingestion suggests that caffeine should be recommended in a customized manner. The development of more precise and individualized guidelines would seem necessary for soccer players.

### 4.5. Strengths, Limitations and Future Lines of Research

The current systematic review presents some limitations related to the different research protocols and performance tests used in the investigations included. Although we selected investigations in which caffeine was compared to an identical situation without caffeine administration, in some investigations, caffeine was co-ingested with other ingredients (e.g., carbohydrates). It is still possible that some of these ingredients produced a synergistic or antagonistic effect on performance. In addition, the dose of caffeine and posology were not uniform among investigations, which could influence some of the outcomes of the research included in the review. In addition, in the investigations included in the analysis there were different competitive levels and age categories while the low number of articles impeded us from knowing if the effect of caffeine on soccer physical performance depends on level or players’ age. Despite these limitations, this review suggests a positive effect of caffeine in increasing soccer players’ physical performance with no or little effect on the levels of muscle damage, perceived effort, or exercising heart rate. Because soccer is a complex sport in which the variables investigated in this systematic review represent only a small proportion of the factors necessary for succeeding, further investigations are necessary to determine the effects of caffeine on more complex and ecological soccer-specific tests, especially involving decision-making situations. In the same way, studies should be undertaken into whether the effect of caffeine is different according to the competitive level or soccer player’s age. 

## 5. Conclusions

In summary, acute caffeine intake of a moderate dose of caffeine before exercise has the capacity to improve several soccer-related abilities and skills such as vertical jump height, repeated sprint ability, running distances during a game and passing accuracy. Likewise, so far, it has been shown that a single and acute dose of caffeine does not have a negative impact on the increase of variables related to muscle damage during official matches. However, more studies are needed to assess whether chronic caffeine consumption could alter muscle damage markers. Moreover, caffeine supplementation does not cause changes in either the perception of effort or heart rate during regular high-intensity intermittent soccer exercises. 

Despite this investigation suggesting several benefits of caffeine in soccer, the use of this stimulant should only be recommended after a careful evaluation of the drawbacks typically associated with the use of caffeine [56]. With this aim, the minimal dose with a positive impact would be recommended (i.e., 3 mg/kg), while it can be consumed in either powder (capsules) or liquid (energy drink or sport drink) forms. Caffeine should only be recommended to athletes who are willing to use ergogenic aids to increase performance and it should be recommended only on an individual basis under careful supervision, in order to avoid the use of this substance in non-responders or athletes who report negative side-effects. Experimenting with caffeine while training, before use in any competition, and avoiding caffeine tolerance may also be further recommendations when using this substance to increase soccer physical performance.

## Figures and Tables

**Figure 1 nutrients-11-00440-f001:**
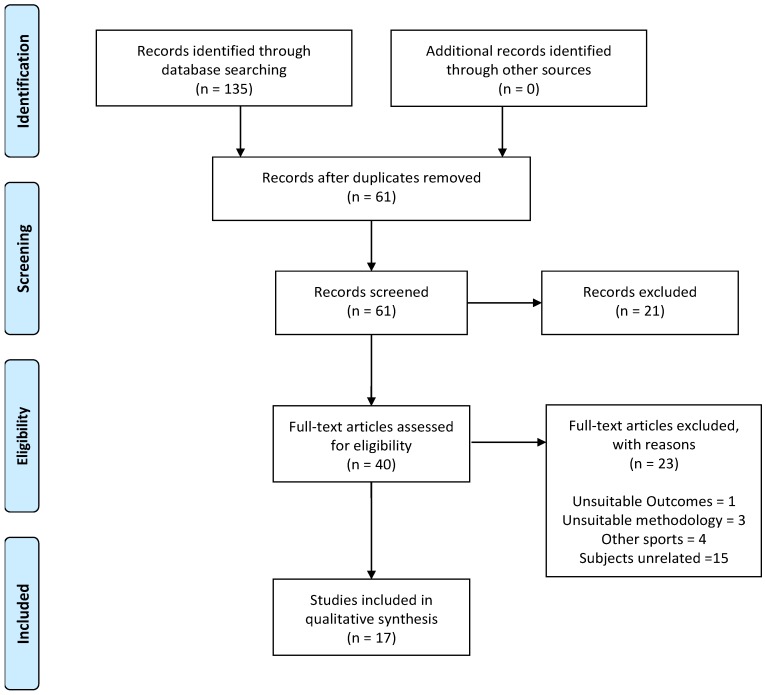
Selection of studies.

**Table 1 nutrients-11-00440-t001:** Summary of studies included in the systematic review that investigated the effect of caffeine ingestion as compared to a placebo on soccer-specific abilities.

Author/s	Population	Intervention	Outcomes Analyzed	Main Conclusion
Ellis M. et al. 2018 [21]	15 male elite youth players (16 ± 1 years)	1, 2 or 3 mg/kg of caffeine capsules 60 min before the start	20-m sprint Arrowhead agility CMJ Yo-Yo IR1	↑ 20-m sprint ↑ Arrowhead agility ↑ CMJ ↑ Yo-Yo IR1
Apostolidis A. et al. 2018 [19]	20 well-trained male players High (*n* = 11) and low (*n* = 9) responders (21.5 ± 4 years)	6 mg/kg of caffeine capsules 60 min before the start	CMJ Reaction time Time to fatigue	↑ CMJ ^†^ Reaction time ↑ Time to fatigue
Guerra MA Jr. et al. 2018 [18]	12 male professional players (23 ± 5 years)	5 mg/kg of caffeine + 20% carbohydrate solution 60 min before the start	CMJ at 1, 3 and 5 min after the conditioning stimulus	↑ CMJ
Ranchordas et al., 2018 [17]	10 male university-standard players (19 ± 1 years)	200 mg (≈2.7 g/kg) of caffeinated gum 5 min before the start	20-m sprint CMJ Yo-Yo IR1	^†^ 20-m sprint ↑ CMJ ↑ Yo-Yo IR1
Andrade Souza, V. et al. 2015 [25]	11 male amateur players (25.4 ± 2.3 years)	6 mg/kg of caffeine capsules 3 h after the LIST	30-m Repeated-Sprint test CMJ LSPT	^†^ 30-m Repeated-sprint test ^†^ CMJ ^†^ LSPT
Jordan, J.et al. 2014 [16]	17 male elite young players (14.1 ± 0.5 years)	6 mg/kg of caffeine capsules 60 min before the start	Sprint time Reaction time	^†^ Sprint time ↑ Reaction time on non-dominant leg
Lara, B. et al. 2014 [27]	18 female semi-professional players (21 ± 2 years)	3 mg/kg of caffeinated energy drinks 60 min before the start	Height and power of jump Average speed of running Total distance covered Number of sprints	↑ Height and power of jump ↑ Average speed of running ↑ Total distance covered ↑Number of sprints
Astorino, T. et al. 2012 [20]	15 female collegiate players (19.5 ± 1.1 years)	255 mL (≈1.3 mg/kg) of caffeinated energy drinks (Redbull) 60 min before the start	Sprint time	^†^ Sprint time
Del Coso, J. et al. 2012 [16]	19 male semi-professional players (21 ± 2 years)	3 mg/kg of caffeine in energy drink 60 min before the start	Maximum height jump Maximum running speed Distance covered Caffeine concentration in urine	↑ Maximum height jump ↑ Maximum running speed ↑Distance covered ↑Caffeine concentrations in urine
Gant, N. et al. 2010 [28]	15 male first team level players (21.3 ± 3 years)	160 mg/L (≈3.7 mg/kg) of caffeinated sport drinks 60 min before the start and every 15 min during the test	Sprint times Jump power Test of passes Blood lactate Post-exercise caffeine in urine	↑ Sprint times ↑ Jump power ^†^ Test of passes ^†^ Blood lactate ↑ Post-exercise caffeine in urine
Foskett et al. 2009 [15]	12 male professional players (23.8 ± 4.5 years)	6 mg/kg of caffeine capsules 60 min before the start	LSPT CMJ	↑ LSPT ↑CMJ
Guttierres, A. P. et al. 2009 [29]	18 male junior players (16.1 ± 0.7 years)	250 mg/L (≈7.2 mg/kg) of caffeinated sport drinks 20 min before and every 15 min during the test	Jump height Illinois agility test	↑ Jump height ^†^ Illinois agility test

↑: statistically significant increase; ^†^ change with no statistical significance; ↓: statistically significant decrease. CMJ: countermovement jump; LIST: Loughborough Intermittent Shuttle Test; LSPT: Loughborough Soccer Passing Test; Yo-Yo IR1: Yo-Yo intermittent recovery test level-1

**Table 2 nutrients-11-00440-t002:** Summary of studies included in the systematic review that investigated the effect of caffeine ingestion as compared to a placebo on serum markers of muscle damage.

Author/s	Population	Intervention	Outcomes Analyzed	Main Conclusion
Guttierres, A. P. et al. 2013 [26]	20 male young players (16.1 ± 0.7 years)	7.2 mg/kg of caffeinated sport drinks 20 min before and every 15 min during the test	Blood glucose Blood lactate Plasma caffeine Free fatty acids Urine caffeine	↑ Blood glucose ↑ Blood lactate ↑ Plasma caffeine ^†^ Free fatty acids ^†^ Urine caffeine
Machado, M. et al. 2010 [30]	15 male players (18.4 ± 0.8 years)	4.5mg/kg of caffeine capsules Immediately before the test	CK LDH ALT AST basophils, eosinophils, neutrophils, monocyte lymphocytes	^†^ CK ^†^ LDH ^†^ ALT ^†^ AST ^†^ basophils, eosinophils, neutrophils, monocyte lymphocytes
Machado, M. et al. 2009 [31]	20 male players (18.8 ± 1 years)	4.5 mg/kg of caffeine capsules Immediately before the test	Basic hemogram CK LDH ALT AST AP γ-GT	^†^ Basic hemogram ^†^ CK ^†^ LDH ^†^ ALT ^†^ AST ^†^ AP ^†^ γ-GT
Machado, M. et al. 2009 [32]	15 male professional players (19 ± 1 years)	5.5 mg/kg of caffeine capsules Immediately before the test	CK LDH ALT AST	^†^ CK ^†^ LDH ^†^ ALT ^†^ AST
Bassini-Cameron, A. et al. 2007 [33]	22 male professional players (26.0 ± 1.6 years)	5 mg/kg of caffeine capsules 60 min before the start	CK LDH ALT AST	↑ CK ^†^ LDH ↑ALT ^†^ AST

↑: statistically significant increase; ^†^ change with no statistical significance; ↓: statistically significant decrease. CK: creatine kinase; LDH: lactate dehydrogenase; ALT: alanine aminotransferase; AST: aspartate aminotransferase; AP: alkaline phosphorylase; γ-GT: γ-glutamyl transferase.

**Table 3 nutrients-11-00440-t003:** Summary of studies included in the systematic review that investigated the effect of caffeine ingestion as compared to a placebo on perceived fatigue and heart rate.

Author/s	Population	Intervention	Outcomes Analyzed	Main Conclusion
Andrade Souza, V. et al. 2015 [25]	11 male amateur players (25.4 ± 2.3 years)	6 mg/kg of caffeine capsules 3 h after the LIST	Perceived effort	^†^ Perceived effort
Jordan, J.et al. 2014 [16]	17 male elite young players (14.1 ± 0.5 years)	6 mg/kg of caffeine capsules 60 min before the start	Heart rate	^†^ Heart rate
Lara, B. et al. 2014 [27]	18 female semi-professional players (21 ± 2 years)	3 mg/kg of caffeinated energy drinks 60 min before the start	Heart rate	^†^ Heart rate
Guttierres, A. P. et al. 2013 [26]	20 male young players (16.1 ± 0.7 years)	7.2 mg/kg of caffeinated sport drinks 20 min before and every 15 min during the test	Perceived effort	^†^ Perceived effort
Astorino, T. et al. 2012 [20]	15 female collegiate players (19.5 ± 1.1 years)	255 mL (≈1.3 mg/kg) of caffeinated energy drinks (Redbull) 60 min before the start	Perceived effort Heart rate	^†^ Perceived effort ^†^ Heart rate
Foskett et al. 2009 [15]	12 male professional players (23.8 ± 4.5 years)	6 mg/kg of caffeine capsules 60 min before the start	Heart rate	^†^ Heart rate

↑: statistically significant increase; ^†^ change with no statistical significance; ↓: statistically significant decrease.

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
