# Peer review of "Caffeine Supplementation and Physical Performance, Muscle Damage and Perception of Fatigue in Soccer Players: A Systematic Review"

_nutrients, 2019, doi:10.3390/nu11020440_

Round 1
Reviewer 1 Report
The manuscript provides an analysis the effect of caffeine intake on soccer physical demands. I would like to commend the authors on providing a well written and interesting analysis. The comments below are provided as a means to enhance the current manuscript.
General comments.
There is a good abstract, but the authors talk about the “several studies have described the effect of caffeine intake on soccer performance” (Line 5) and the literature really talk about soccer demands or physical demands or physical performance, but not about soccer performance. The soccer performance is multifactorial. Also, this idea is present in several parts of the manuscript.
There is a good analysis about the situation in the introduction, but I believe the manuscript could be enhanced. In this way, I recommend to rewrite the “endurance capacity is also essential during a football match”, the idea of endurance capacity is certainly obsolete, since football is understood today as a sport with intermittent efforts where high intensity are determinant, so that part should be rewritten.
The method is the part that worries me most about the entire manuscript. As a general comment, I can see a great inconsistency between the title and the method that the authors use. It is confusing because of the terms “systematic review” or “meta-analysis”. If you want to see the “effect” of something, it should appear statistical analysis. If not, I would recommend changing the title and remove the word “effect”.
Line 82. It puts “meta-analysis”. Is it a meta-analysis or a systematic review? Inconsistency. In the title appears “systematic review”
Lines 85-85. Why did the authors choose the period between 2007 and 2018? Justify. Are there other literature reviews published previously?
Lines 93-94. What kind of disagreements were resolved?
Line 105. Inclusion criteria. “Published in any language”. Perhaps, the articles published in a language different from English not ensure enough quality in the papers added in the review.
Lines 115-122. It is highly recommendable to use the PICO strategy to classify the articles: Participants, Intervention(s), Comparator, Outcomes. The author did not follow this strategy included in the PRISMA guidelines.
Line 117. There is a red comma.
Line 125. Did you find just 93 articles in the first search? It is necessary to show all the results that you found in the first search. Maybe 1000, 2000 or 3000.
Figure 1. “n=8”. What were the “other sources”?
Figure 1. It is necessary to show the reasons for excluding the articles. It is true that you have explained the exclusion criteria, but it should be showed how many articles were excluded and the reasons of all of them.
Figure 1. It puts “meta-analysis”. Is it a meta-analysis or a systematic review? Inconsistency. In the title appears “systematic review”
It does not appear the data analysis. How were the statistic calculated? Did you use a specific programme like, of example, “Comprehensive meta-analysis”? or just SPSS? Did you calculate the effect sizes? d Cohen? Explain and justify.
Results.
The results are mixed high performance studies in soccer, with studies on amateur, youth or women soccer players. In this sense, the presentation of these results should be better refined, since the effects of caffeine are not the same on an adolescent as on an adult, nor on a professional as on an amateur. This part should be rewritten.
Discussion and Conclusions.
There is some good discussion and conclusion.
Author Response
Dear Journal Editor and Reviewer,
We appreciate again the time you devoted to reading our manuscript and helping us to craft an improved version. We are pleased to clarify your concerns which we believe have improved the quality and applicability of your work. Please, find below our response to your observations. We have made a concerted attempt to systematically address the specific concerns raised for this revision and we have highlighted the alterations to this revision within the manuscript in green for your convenience.
Reviewer 1
The manuscript provides an analysis the effect of caffeine intake on soccer physical demands. I would like to commend the authors on providing a well written and interesting analysis. The comments below are provided as a means to enhance the current manuscript.
General comments.
1. There is a good abstract, but the authors talk about the “several studies have described the effect of caffeine intake on soccer performance” (Line 5) and the literature really talk about soccer demands or physical demands or physical performance, but not about soccer performance. The soccer performance is multifactorial. Also, this idea is present in several parts of the manuscript.
Answer: Thank you for your observation. Authors have changed soccer performance by soccer physical performance throughout the text.
2. There is a good analysis about the situation in the introduction, but I believe the manuscript could be enhanced. In this way, I recommend to rewrite the “endurance capacity is also essential during a football match”, the idea of endurance capacity is certainly obsolete, since football is understood today as a sport with intermittent efforts where high intensity are determinant, so that part should be rewritten.
Answer: Thank you for your observation. The authors have deleted that phrase in order to avoid any misunderstanding given that as you tell us soccer requires intermittent efforts.
3. The method is the part that worries me most about the entire manuscript. As a general comment, I can see a great inconsistency between the title and the method that the authors use. It is confusing because of the terms “systematic review” or “meta-analysis”. If you want to see the “effect” of something, it should appear statistical analysis. If not, I would recommend changing the title and remove the word “effect”.
Answer: Thanks for your observation. The authors have eliminated the word effect from the title of the manuscript and now reads: “Caffeine supplementation and physical performance, muscle damage and perception of fatigue in soccer players: a systematic review.”
4. Line 82. It puts “meta-analysis”. Is it a meta-analysis or a systematic review? Inconsistency. In the title appears “systematic review”
Answer: Thank you for your comment. In order to be fully clarified, the authors express that the present manuscript is a systematic review with no subsequent meta-analysis. For its elaboration, the authors have followed the guidelines of PRISMA whose nomenclature is “Preferred Reporting Items for Systematic Review and Meta-Analyses”. This protocol is used to perform both systematic reviews and systematic reviews with meta-analysis.
5. Lines 85-85. Why did the authors choose the period between 2007 and 2018? Justify. Are there other literature reviews published previously?
Answer: Thanks for your interest. In order to explain why this date was used, the authors have added the following: The search covered from January July 2006, when Hespel et al., in July 2006 [23] suggested the use of caffeine as an effective supplement for soccer athletic performance, to November 2018.
6. What kind of disagreements were resolved?
Answer: Thank you for this comment. The extraction of data and the evaluation of each research’s quality is not a totally objective process because, sometimes, the information is not clearly presented in the manuscript (for example, the level of participants, the type of experiment, pre-exercise standardizations, etc). In the case that there were disagreements between the two authors (JMA and JCG) who performed the data extraction, they reached to a conclusion through personal discussion.
7. Line 105. Inclusion criteria. “Published in any language”. Perhaps, the articles published in a language different from English not ensure enough quality in the papers added in the review.
Answer: Thank you for your comment. The authors want to indicate that the revised databases presented some articles in languages other than English. In this sense, articles were reviewed in English, German, French, Italian, Spanish and Portuguese in a attempt to increase the number of manuscripts included in the review. At the end, out of the 17 articles included in the review, 16 are in English and 1 in Portuguese which confirm the high quality of them. Because of this, authors have added the languages of the articles they reviewed in the inclusion criteria: “8) the languages were restricted to English, German, French, Italian, Spanish and Portuguese”.
8. Lines 115-122. It is highly recommendable to use the PICO strategy to classify the articles: Participants, Intervention(s), Comparator, Outcomes. The author did not follow this strategy included in the PRISMA guidelines.
Answer: Thanks for your observation. The authors have included in the Search Strategies section the following paragraph in order to explain the PICOS strategy: “The present article is a systematic review focusing on the impact of caffeine intake on soccer physical performance and it was conducted following the Preferred Reporting Items for Systematic Review and Meta-Analyses (PRISMA) guidelines and the PICOS model for the definition of the inclusion criteria: P (Population): “soccer players”, I (Intervention): “impact of caffeine on soccer physical performance, muscle damage and perception of fatigue”, C (Comparators): “same conditions with placebo”, O (Outcome): “soccer-specific abilities, serum markers of muscle damage and perceived fatigue (RPE) and heart rate”, and S (study design): “double-blind and randomized cross-over design” [22].”
Likewise, the authors have modified the identification of the columns of the tables to conform to the PICOS strategy.
9. Line 117. There is a red comma.
Answer: Thanks for your observation. The authors have changed the red comma to a black comma.
10. Line 125. Did you find just 93 articles in the first search? It is necessary to show all the results that you found in the first search. Maybe 1000, 2000 or 3000.
Answer: Thanks for your observation. The authors after reviewing both databases obtained a total of 135 references related to the search strategy. This fact has been modified in figure 1 and indicated in the text.
11. Figure 1. “n=8”. What were the “other sources”?
Answer: Thank you for your interest. After reviewing the data bases the authors have realized that within the database of Web Of Science (WOS) are other databases such as BCI, BIOSIS, CCC, DIIDW, INSPEC, KJD, MEDLINE, RSCI, SCIELO. The 8 articles previously indicated as obtained from other sources belonged to these databases that have been included in Records identified through database searching.
12. Figure 1. It is necessary to show the reasons for excluding the articles. It is true that you have explained the exclusion criteria, but it should be showed how many articles were excluded and the reasons of all of them.
Answer: Thanks for your observation. Authors added in the figure 1 the reasons for excluding the articles.
13. Figure 1. It puts “meta-analysis”. Is it a meta-analysis or a systematic review? Inconsistency. In the title appears “systematic review”
Answer: Thanks for your observation. The authors have eliminated the Studies included in qualitative synthesis (meta-analysis) section in Figure 1.
14. It does not appear the data analysis. How were the statistic calculated? Did you use a specific programme like, of example, “Comprehensive meta-analysis”? or just SPSS? Did you calculate the effect sizes? d Cohen? Explain and justify.
Answer: Thanks for your observation. As we have mentioned before, there was no subsequent meta-analysis after a systematic review. Thus, no statistical analysis was carried out.
Results.
15. The results are mixed high performance studies in soccer, with studies on amateur, youth or women soccer players. In this sense, the presentation of these results should be better refined, since the effects of caffeine are not the same on an adolescent as on an adult, nor on a professional as on an amateur. This part should be rewritten.
Answer: We appreciate your comment. The small number of articles obtained in relation to the objective of the systematic review does not allow us to determine if the impact of caffeine on the analyzed outcomes depends on the competitive level or age. We have acknowledged this as a limitation: In addition, the investigations included in the analysis there were different competitive levels and ages categories while the low number of articles impeded us to know if the effect of caffeine on soccer physical performance depends on level or players’ age.
Discussion and Conclusions.
There is some good discussion and conclusion.
Answer: On behalf of all co-authors, many thanks for the insightful comments and suggestions for this review.

Reviewer 2 Report
- Keywords, please use different terms not included in the title such as football, RPE, etc.
- L47, L55, L60, L65, L69, L72, L84…please delete the extra space through the text.
- L54, please replace “the margin between winning and losing teams can be small” with “the margins of victory are close/reduced”.
- L111, please write into brackets the key factors assessed with the PEDro instrument.
- L112, please add points to the range values for a better understanding.
- L114, please write into brackets: (<4 points).
- L120-123, please justify with references the use of these cluster of the topic.
- L126, please write (see Figure 1).
- Tables 1-3, please write MAIN CONCLUSION instead of CONCLUSION.
- Table 1, sample of the Apostolidis’ study, reduce the spaces when describing the n.
- Tables 1-3, please the arrows should be better justified in the legend, please use statistically significant increase/ decrease.
- Discussion, conclusions and limitations are well written and presented. However, I would like to read a further research focused on the experimental approaches that should be done in a future to continue testing the effects of caffeine on soccer players.
Author Response
Dear Journal Editor and Reviewer,
We appreciate again the time you devoted to reading our manuscript and helping us to craft an improved version. We are pleased to clarify your concerns which we believe have improved the quality and applicability of your work. Please, find below our response to your observations. We have made a concerted attempt to systematically address the specific concerns raised for this revision and we have highlighted the alterations to this revision within the manuscript in green for your convenience.
Reviewer 2
1. -Keywords, please use different terms not included in the title such as football, RPE, etc.
Answer: Thanks for your recommendation. Authors have modified the list of keywords and not it reads: football; RPE; DOMS; sport performance; supplementation; ergogenic aids
2. L47, L55, L60, L65, L69, L72, L84…please delete the extra space through the text.
Answer: Thanks for your appreciation. After reviewing the text, the authors have erased the double space observed in different parts of it.
3. L54, please replace “the margin between winning and losing teams can be small” with “the margins of victory are close/reduced”.
Answer: Thanks for your recommendation. Authors have made the change proposed by the reviewer.
4. L111, please write into brackets the key factors assessed with the PEDro instrument.
Answer: Thanks for the recommendation. Authors have added a phrase indicating the key factors assessed with the PEDro instrument: “There were no filters applied to the soccer players’ level, sex or age to increase the power of the analysis. Moreover, the Physiotherapy Evidence Database scale (PEDro), which key factors assessed eligibility criteria, random allocation, baseline values, success of the blinding procedures, power of the key outcomes, correct statistical analysis and measurement of participants’ distribution of studies, was used to evaluate whether the selected randomized controlled trials were scientifically sound: 9–10 = excellent, 6–8 = good, 4–5 = fair, and<4 = poor) [24].”
5. L112, please add points to the range values for a better understanding.
Answer: Thanks for the comment. Authors have added points to the range values for a better understanding.
6. L114, please write into brackets: (<4 points).
Answer: Thanks for the observation. Authors have written into brackets: (<4 points).
7. L120-123, please justify with references the use of these cluster of the topic.
Answer: Thanks for the observation. Authors have justified that paragraph with a reference where indicate the importance of these clusters in soccer performance. Concretely, authors have included this phrase and reference: “because of its importance in soccer performance [4].”
8. L126, please write (see Figure 1).
Answer: Thanks for the observation. Authors have added (see Figure 1).
9. Tables 1-3, please write MAIN CONCLUSION instead of CONCLUSION.
Answer: Thanks for the observation. Authors have changed MAIN CONCLUSION instead of CONCLUSION.
10. Table 1, sample of the Apostolidis’ study, reduce the spaces when describing the n.
Answer: Thanks for the observation. Authors have reduced the spaces when describing the n.
11. Tables 1-3, please the arrows should be better justified in the legend, please use statistically significant increase/ decrease.
Answer: Thanks for the observation. Authors have justified in the legend the arrows using use statistically significant increase/ decrease.
12. Discussion, conclusions and limitations are well written and presented. However, I would like to read a further research focused on the experimental approaches that should be done in a future to continue testing the effects of caffeine on soccer players.
Answer: Thanks for the recommendation. Authors have included in the limitation section the next paragraph: “Because soccer is a complex sport in which the variables investigated in this systematic review represent only a small proportion of the factors necessary for succeeding, further investigations are necessary to determine the effects of caffeine on more complex and ecological soccer-specific tests, especially involving decision-making situations. In the same way, studies should be proposed to know if the effect of caffeine is different according to the competitive level or soccer players’ age.”

Round 2
Reviewer 1 Report
The authors have improve the manuscript, so it could be published in this format.